# Effects of Organic Fertilizer Application on Tomato Yield and Quality: A Meta-Analysis

**Fucheng Gao [1], Haijun Li [1], Xiaoguo Mu [1], Hu Gao [1], Ying Zhang [1], Ruimiao Li [1], Kai Cao [2,\*] and Lin Ye [1,\*]**

[1] College of Agriculture, Ningxia University, Helanshan Xilu No. 489, Yinchuan 750021, China
[2] The Agriculture Ministry Key Laboratory of Agricultural Engineering in the Middle and Lower Reaches of Yangtze River, Institute of Agricultural Facilities and Equipment, Jiangsu Academy of Agricultural Sciences, Nanjing 210000, China
\* Correspondence: kcao@jaas.ac.cn (K.C.); yelin.3993@163.com (L.Y.)

**Abstract:** Tomatoes are a globally cultivated and popular vegetable. The output and quality of tomatoes are significantly influenced by the use of organic fertilizers. It was discovered that organic fertilizers increase tomato productivity and improve fruit quality. The influence of organic fertilizers on tomato yield and quality is shown to be complex and dependent on soil organic matter, total soil nitrogen, organic fertilizers kinds, and other variables. In this review paper, we evaluated 769 data sets from 107 research papers and determined that organic fertilizers can enhance the tomato yield by 42.18%. Compared to the control group, soluble solids, soluble sugar, lycopene, vitamin C, and nitrate were raised by 11.86%, 42.18%, 23.95%, 18.97%, and 8.36%, respectively. In general, the soil organic matter >20 g·kg$^{-1}$ and organic fertilizers significantly improved the tomato sugar/acid content ratio and VC, whereas under total soil nitrogen >1 g·kg$^{-1}$, organic fertilizers had significant differences in tomato soluble solids, soluble sugar, lycopene, and vitamin C, with different organic-fertilizer types having different effects on tomato quality. When comparing animal and plant organic fertilizers to other forms of organic fertilizers, we observed that tomato quality varied significantly. We also evaluated the impact of different cultivation methods, soil organic matter, total soil nitrogen, soil pH, and types of organic fertilizers on the tomato yield and quality. The results gave valuable information and direction for the use of organic fertilizers in greenhouse production.

**Keywords:** organic fertilization; lycopene; nitrogen; organic matter; meta-analysis





## 1. Introduction

It is well-known that a healthy diet is essential for preventing chronic diseases such as cancer, cardiovascular disease, cognitive function, and osteoporosis, as well as improving antioxidant levels and controlling body weight [1]. Lower serum or plasma lycopene levels associated with increased cancer risk, tomato, and its derivatives have not only a high nutritional value but also antioxidant [2,3], anti-inflammatory, and anticancer properties [4]. Furthermore, the tomato is one of the most widely cultivated vegetables [5]. More than 180 million tons of tomatoes are produced worldwide, making them essential for a healthy and balanced diet due to their functional compounds, such as lycopene, vitamins, minerals, and proteins [6].

The application of chemical fertilizers is currently one of the most commonly used methods in intensive agriculture [7,8]. However, the long-term application of chemical fertilizers can cause many negative effects. For example, most of the nutrients added to the soil are not absorbed by plants. Studies have shown that more than 50% of the nitrogen and 90% of the phosphorus in chemical fertilizers are lost to the atmosphere or water sources [9], resulting in greenhouse gas emissions, water eutrophication, and other environmental issues [10–12]. Furthermore, excessive chemical fertilizer application can result in decreased food safety and lower vegetable quality, such as nitrate accumulation in

plants [13]. Nowadays, using organic fertilizers is an efficient method to achieve sustainable agricultural development. The nutrient release rate of organic fertilizers is slow and hardly exceeds the absorption capacity of plants compared with chemical fertilizers [14]. Organic fertilizers have a low nutrient content, and their nutrient release rate depends on water and temperature conditions of the soil. The application of organic fertilizers not only improves soil physical and chemical properties [15,16], soil fertility, and soil water storage capacity [17,18], but it also can promote vegetative and reproductive plant growth effectively [19,20], thereby improving plant quality [21–23]. It has been reported that the application of chicken manure increased tomato yield and quality by 43% and 23%, respectively, especially the soluble protein by 124% and the titratable acid by 118% [24]. Therefore, using organic fertilizers can improve fertilizers' utilization rates.

However, many studies have been conducted on the qualitative analysis of tomato yield and quality, and only a few studies have investigated the quantitative effects of organic fertilizers on tomato yield and quality under different soil organic matter and soil total nitrogen. This paper conducted a comprehensive quantitative analysis of tomato yield and quality, with inconsistent results through meta-analysis. It provided evidence for the application of organic fertilizers in tomato planting.

## 2. Materials and Methods

### 2.1. Data Collection

A complete search of the literature was undertaken on ISI Web of Science, China National Knowledge Internet, and Google academic to collect data for a study on the effects of organic fertilizers on tomato output and quality published as of April 2022. The literature search was conducted with "tomato" and "organic fertilizer" as the main keywords. The following criteria were used to select suitable studies: (1) the study must include organic-fertilizer treatment and a control with no fertilizer treatment; (2) it must describe the test site, cultivation type, and organic-fertilizer type; and (3) it must provide the mean and standard deviation of the treatment group and the control group, and N (number of replicates). If the standard error (SE) was given but the SD was not, we converted the SE to SD by using the formula $SD = SE \times \sqrt{n}$ [25].

We divided tomato-quality variables into total soluble solids (TSS), soluble sugar (SS), sugar/acid content ratio (SAR), lycopene, vitamin C (VC), and nitrates. We collected soil organic matter, soil total nitrogen, soil pH, organic-fertilizer type, and cultivation type as subgroups. Since the sample size of the cultivation type and soil pH was insufficient for grouping, only soil organic matter was divided into two groups: (1) SOM < 20 g·kg$^{-1}$ and (2) SOM ≥ 20 g·kg$^{-1}$. Soil total nitrogen was divided into three groups: (1) TN < 0.1 g·kg$^{-1}$, (2) 0.1 g·kg$^{-1}$ < TN < 1 g·kg$^{-1}$, and (3) TN > 1 g·kg$^{-1}$. The organic-fertilizer categories are divided into four groups: (1) animal (insert space before animal) organic fertilizer, (2) plant organic fertilizer, (3) mixed organic fertilizer, and (4) other organic fertilizer (biogas slurry, kitchen waste, etc.)

Through searching and screening, 107 papers and 769 data pairs were analyzed to determine the effect of organic fertilizers on tomato yield and quality. The distribution of the experimental sites is displayed in Figure 1.

### 2.2. Data Analysis

The meta-analysis was conducted in Metawin 2.1 (Miami, FL, USA) and GraphPad 8.0 software (San Diego, CA, USA). Using the response ratio (R) [26] as a measure of effect size, each group of data was calculated using the following formula:

$$R = \frac{X_t}{X_c} \tag{1}$$

$$\ln R = \ln\left(\frac{X_t}{X_c}\right) \tag{2}$$

where $t$ deals with the average of the set of variables, $c$ is the average of the control variables, $\ln R$ is the logarithmic response, and the variance calculation formula is calculated as follows:

$$v_{\ln R} = \frac{SD_t^2}{n_t\, X_t} + \frac{SD_c^2}{n_c\, X_c} \qquad (3)$$

where $SD$ is the standard error and $n$ is the number of repetitions. To facilitate interpretation, the formula percentage change $= (R - 1) \times 100\%$ was used.

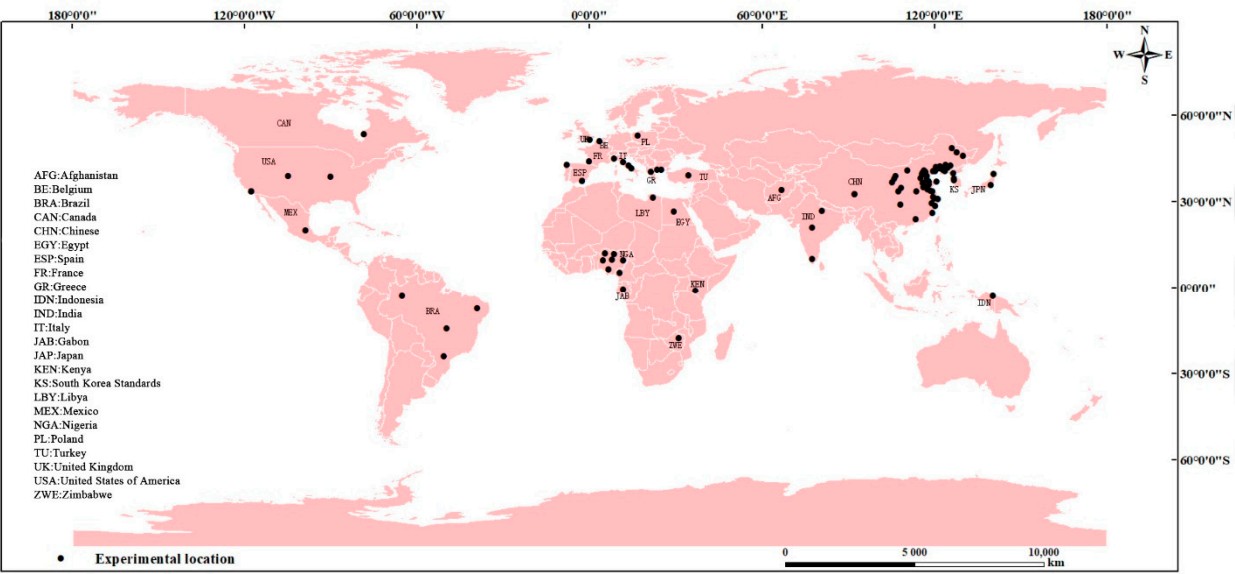

**Figure 1.** Geographic distribution of the experimental sites in this meta-analysis.

Higgins $I^2$ statistic and Q test [27] were used to assess the degree of heterogeneity and indicate differences between studies. Fixed-effect models were used for studies with low heterogeneity ($I^2 \leq 50\%$ and $p > 0.05$), and random-effect models were used for studies with opposite heterogeneity ($I^2 > 50\%$ or $p < 0.05$). A forest map is produced from the pooled results. In the forest plot, the mean ratio response and 95% confidence interval (CI) are shown as dots and bars, respectively. We used Egger tests [26] and Rosenthal's fail-safe number, N, to assess publication bias. Rosenthal's fail-safe numbers of tomato yield, soluble solids, soluble sugars, sugar–acid ratio, lycopene, vitamin C, and nitrate were 48,086,680, 21,344, 190,874, 15,074, 619,422, 223,898, and 1153, respectively (Table 1), which were significantly higher than $N_f$ ($5 \times 350 + 10$, $5 \times 95 + 10$, $5 \times 67 + 10$, $5 \times 51 + 10$, $5 \times 47 + 10$, $5 \times 119 + 10$, and $5 \times 40 + 10$, respectively), indicating that the result is reliable.

**Table 1.** Exploring potential publication bias and robustness by Egger tests and Rosenberg's fail-safe numbers ($N_f$).

| Yield and Quality | n | Egger Tests | $N_f$ |
|---|---|---|---|
| Yield | 350 | 0.3169 | 48,086,680 |
| TSS | 95 | 0.1412 | 21,344 |
| SS | 67 | 0.2267 | 190,874 |
| SAR | 51 | 0.4486 | 15,074 |
| Lycopene | 47 | 0.2413 | 619,422 |
| VC | 119 | 0.2101 | 223,898 |
| Nitrate | 40 | 0.2392 | 1153 |

These showed that, with $N_f > 5 \times n + 10$, there is no potential bias in the total effect sizes for yield, TSS, SS, SAR, lycopene, VC, and nitrate ($p > 0.05$).

## 3. Results

### 3.1. Overall Effect of Organic Fertilizer on Tomato Yield and Quality

The overall effect of organic fertilizers on tomato yield and quality variables (Figure 2). Specifically, increased organic fertilizers significantly improved the tomato yield by 42.18% (95% CI: 36.22% to 48.38%). For tomato quality, increased organic fertilizers significantly improved TSS by 11.86% (95% CI: 7.55% to 16.33%), SS by 42.18% (95% CI: 25.51% to 61.06%), and SAR by 6.17% (95% CI: −8.9% to 23.75%). SAR indices of tomatoes treated with organic fertilizers did not change. Organic fertilizers significantly increased lycopene by 23.95% (95% CI: 10.65% to 38.86%), VC by 18.97% (95% CI: 15.26% to 22.80%), and nitrate by 8.36% (95% CI: 1.52% to 15.66%), compared to the no-fertilizer application. The regression analysis revealed that tomato VC was linearly and positively correlated with soil organic matter, while TSS, SS, SAR, and nitrate were non-linearly correlated. Similarly, tomato vitamin C was linearly and significantly positively correlated with total soil nitrogen, while TSS, SS, SAR, and nitrate were non-linearly correlated with total soil nitrogen (Figure 3).

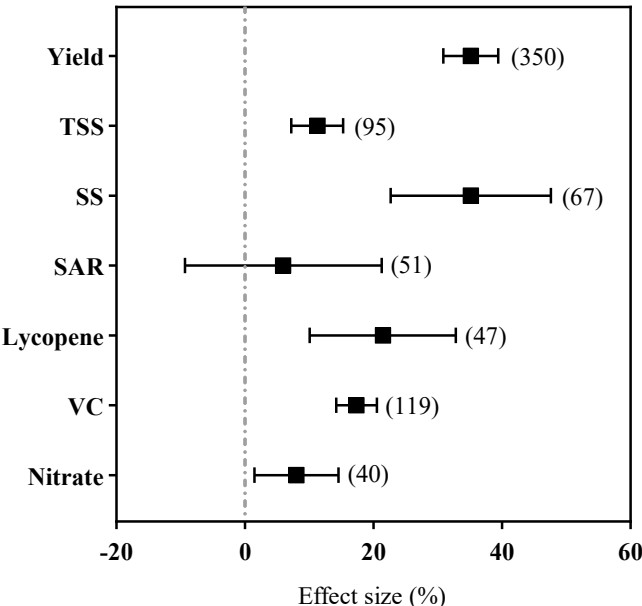

**Figure 2.** Organic fertilizers' effect on tomato yield and quality. Points represent the mean effect, while bars represent 95% confidence intervals. The numbers at the right side of the confidence intervals represent the sample sizes. TSS, total soluble solids; SS, soluble sugar; SAR, sugar/acid content ratio; VC, vitamin C.

### 3.2. Effect of Organic Fertilizer Combined with Soil Organic Matter, Total Soil Nitrogen, and Type of Organic Fertilizer on Tomato Yield

It can be concluded that when soil organic matter is <20 g·kg$^{-1}$ (95% CI: 28.15% to 45.99%) or ≥20 g·kg$^{-1}$ (95% CI: 28.98% to 52.49%), the application of organic fertilizers has no effect on the tomato yield (Figure 4a). In terms of total soil nitrogen, for total soil nitrogen <0.1 g·kg$^{-1}$, 0.1–1 g·kg$^{-1}$, and >1 g·kg$^{-1}$, the tomato yield increased by 73.64, 40.96, and 38.69%, respectively (Figure 4b). Subgroups were made based on the type of organic fertilizers, and a significant increase in tomato yield was observed for each type (Figure 4c). Animal organic fertilizer improved the yield by 50.53%, organic plant fertilizer by 35.78%, animal-and-plant mixed organic fertilizers by 37.16%, and other organic-fertilizer types by 31.98%. Except for animal organic fertilizer, there was no significant difference in the tomato yield among the other three types of organic fertilizers, and the 95% confidence intervals for each group overlapped. The soil pH was divided into two subgroups (Figure 4d). When the soil pH was ≥7, the effect of organic fertilizers on the tomato yield was 33.04% compared with that of no fertilizers. When the soil pH was <7, the effect value of increasing organic

fertilizers on the tomato yield was 60.06%, indicating a significant yield. Depending on the cultivation type, the effect of applying organic fertilizers on the yield in field experiments was 44.26%, compared to 27.26% in the greenhouse (Figure 4e).

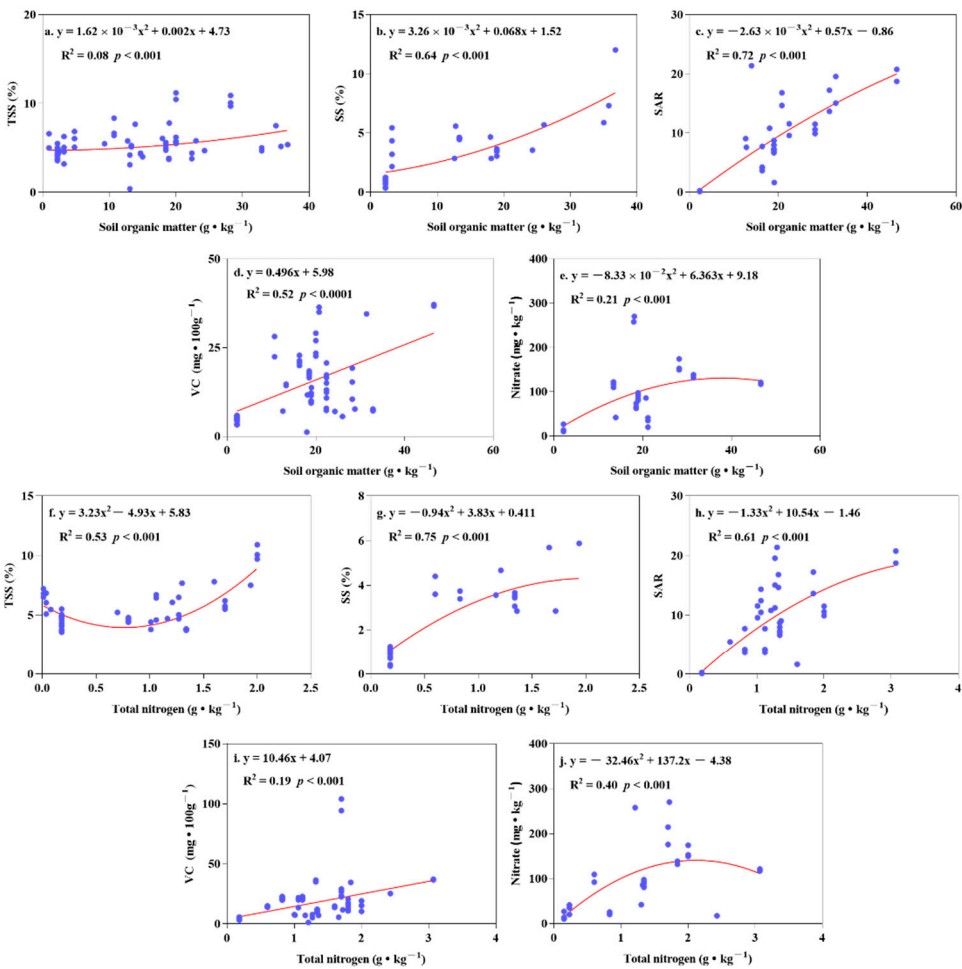

**Figure 3.** Relationship between soil organic matter, total soil nitrogen, and tomato quality: TSS (**a**), SS (**b**), SAR (**c**), VC (**d**), and nitrate (**e**); TSS (**f**), SS (**g**), SAR (**h**), VC (**i**), and nitrate (**j**).

### 3.3. Effect of Organic Fertilizer Combined with Soil Organic Matter, Total Soil Nitrogen, and Organic-Fertilizer Type on TSS

When the soil-organic-matter content was $\geq 20$ g·kg$^{-1}$ and $<20$ g·kg$^{-1}$, the tomato's soluble solid content increased by 14.17% and 11.47%, respectively (Figure 5a). For the total soil nitrogen content, when the total soil nitrogen was $>1$g·kg$^{-1}$ and 0.1–1 g·kg$^{-1}$, the tomato's soluble solids increased by 13.89% and 7.66%, while $<0.1$ g·kg$^{-1}$ had no change in soluble solids (Figure 5b). With the increase in soil total nitrogen content, the effect of organic fertilizers on improving tomato soluble solids tended to be positive and significant. Different organic fertilizers had different effects in regard to improving tomato soluble solids (Figure 5c). The percentage increases were 14.21%, 11.24%, and 8.43% in plant organic fertilizer, animal organic fertilizer, and mixed organic fertilizer, respectively. Other organic fertilizer had no effect, which was 5.73%. Plant organic fertilizer significantly improved tomato soluble solids. At a pH < 7, the percentage increase of TSS was 16.67%, whereas at a pH > 7, the percentage increase was 12.52% (Figure 5d). In conclusion, the application of organic fertilizers on acidic soil significantly improved the tomato soluble solids more than that on alkaline soil (Figure 5e). The soluble solid content of tomatoes cultivated in a greenhouse is higher (16.81%) than that cultivated in the field (4.98%). There is a statistical difference between them.

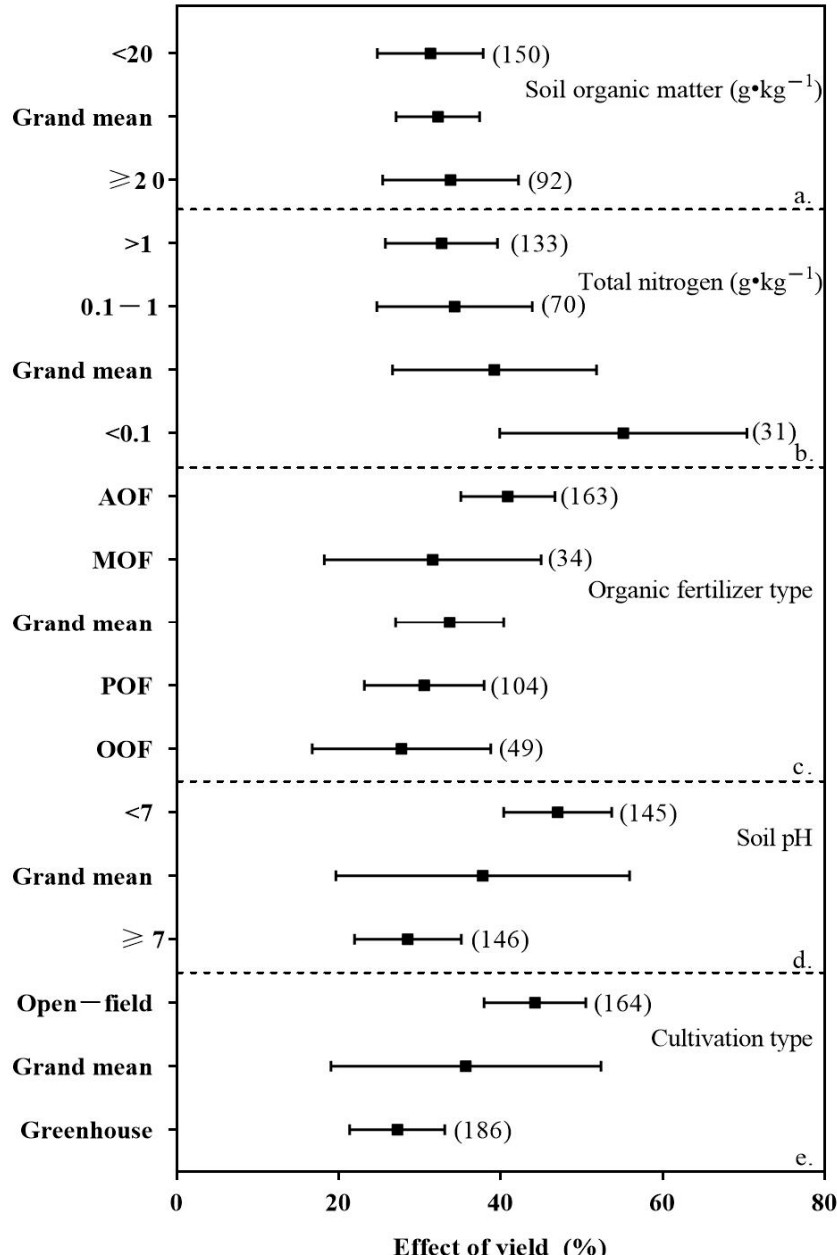

**Figure 4.** Organic fertilizers combined with soil organic matter (**a**), total soil nitrogen (**b**), type of organic fertilizers (**c**), soil pH (**d**), and cultivation type (**e**) on tomato yield percentage changes. Points represent the mean effect, while bars represent 95% confidence intervals. The numbers on the right side of the confidence intervals represent the sample sizes. AOF, animal organic fertilizer; MOF, mixed animal-and-plant organic fertilizer; POF, plant organic fertilizer; OOF, other organic fertilizer.

### 3.4. Effect of Organic Fertilizer Combined with Soil Organic Matter, Total Soil Nitrogen, and Organic-Fertilizer Type on SS and SAR

The percentage change of organic fertilizers on the soluble sugar of tomato increased significantly when the organic matter was <20 g·kg$^{-1}$; the percentage change of organic fertilizers on the soluble sugar of tomato increased by 60.35% (Figure 6a). The percentage change was −18.09% when the organic fertilizers were ≥20 g·kg$^{-1}$. When the total soil nitrogen was 0.1–1 g·kg$^{-1}$ and >1 g·kg$^{-1}$, organic fertilizers had positive effects on tomato soluble sugar, and the percentage changes were 51.86% and 30.53%, respectively (Figure 6b). Both plant organic fertilizers and animal organic fertilizers had positive effects on tomato soluble sugar, and the respective percentage changes were 48.04% and 48.02%; thus, the

confidence intervals overlap (Figure 6c). However, there was no significant difference between mixed organic and other types of organic fertilizers in the percentage changes in tomato soluble sugar. The confidence intervals overlapped, and percentage changes were 25.03% and 18.14%. When the pH was ≥7, organic fertilizers increased tomato soluble sugar with a percentage change of 20.86%, but there was no significant difference (Figure 6d). The increase of soluble sugar in tomatoes with a pH < 7 was significantly different, with a percentage change of 84.52%. Both cultivation types had positive effects on the soluble sugars. The application of organic fertilizers in facility cultivation had significant effects on tomato soluble sugar, with a percentage change of 46.13%, but no significant effect was observed in the open field, with a percentage change of 29.82% (95% CI: −3.43% to 74.53%) (Figure 6e). As can be seen from the figure, compared with no application of organic fertilizers, the cultivation type, organic-fertilizer type, soil organic matter, soil pH, and total soil nitrogen had no significant influence on the tomato sugar/acid ratio under the increased application of organic fertilizers compared to no application of organic fertilizers (Figure 7).

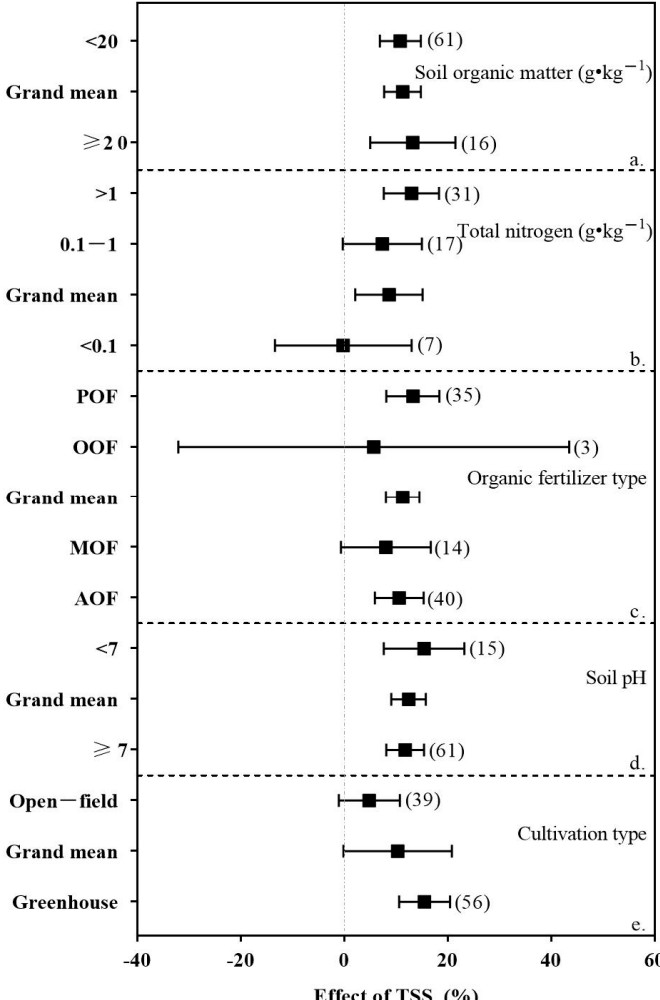

**Figure 5.** Organic fertilizers combined with soil organic matter (**a**), total soil nitrogen (**b**), type of organic fertilizers (**c**), soil pH (**d**), and cultivation type (**e**) on tomato TSS (total soluble solids) percentage changes. Points represent the mean effect, while bars represent 95% confidence intervals. The numbers on the right side of the confidence intervals represent the sample sizes. AOF, animal organic fertilizer; MOF, mixed animal-and-plant organic fertilizer; POF, plant organic fertilizer; OOF, other organic fertilizer.

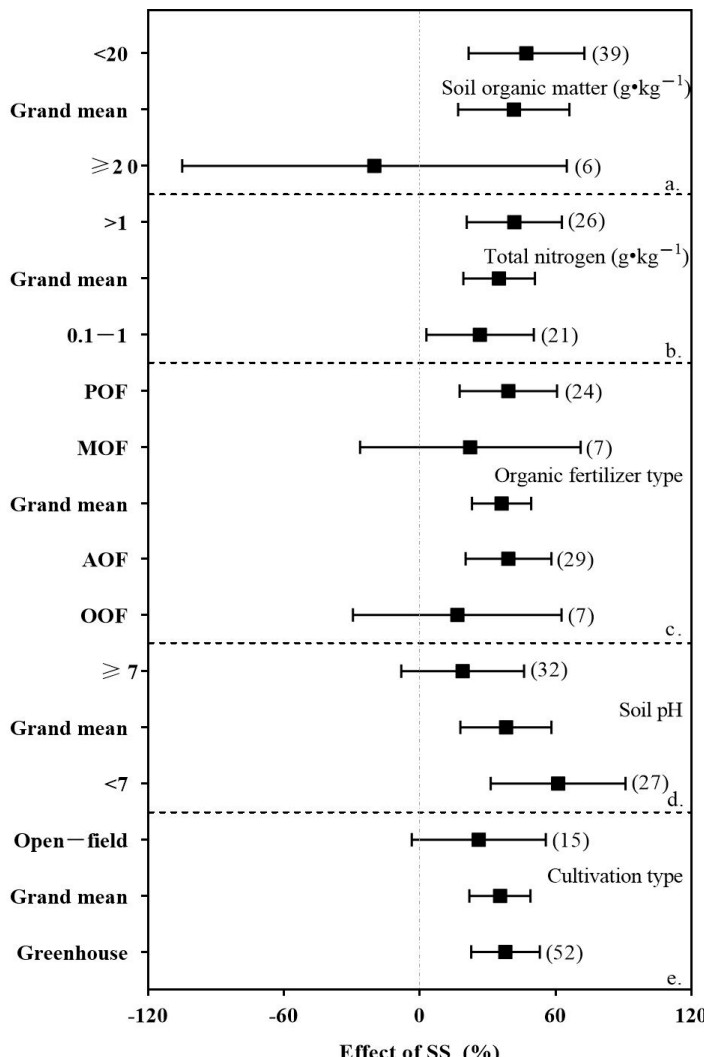

**Figure 6.** Organic fertilizers combined with soil organic matter (**a**), total soil nitrogen (**b**), type of organic fertilizers (**c**), soil pH (**d**), and cultivation type (**e**) on tomato SS (soluble sugar) percentage changes. Points represent the mean effect, while bars represent 95% confidence intervals. The numbers on the right side of the confidence intervals represent the sample sizes. AOF, animal organic fertilizer; MOF, mixed animal-and-plant organic fertilizer; POF, plant organic fertilizer; OOF, other organic fertilizer.

### 3.5. Effect of Organic Fertilizer Combined with Soil Organic Matter, Total Soil Nitrogen, and Organic-Fertilizer Type on Lycopene

The soil organic matter had no significant effect on tomato lycopene under organic fertilizers (Figure 8a). When the total soil nitrogen was >1 g·kg$^{-1}$, the lycopene increased by 29.67%, while when it was <0.1 g·kg$^{-1}$, there was no statistically significant difference (Figure 8b). The plant organic fertilizer significantly improved tomato lycopene, with a percentage change of 35.38% (Figure 8c). Animal organic fertilizer and other types had no significant effect on tomato lycopene improvement, with percentage changes of 25.60% and 10.56%, respectively. The soil pH had a significant effect on lycopene under organic fertilizers (Figure 8d). When the pH was < 7, the percentage change of adding organic fertilizers to lycopene was 18.42%; at a pH ≥ 7, the percentage change was 28.30%. The improvement effect on lycopene in tomatoes cultivated in the open field was significantly higher (27.58%) than that in facility cultivation (14.09%) under organic fertilizers' application (Figure 8e).

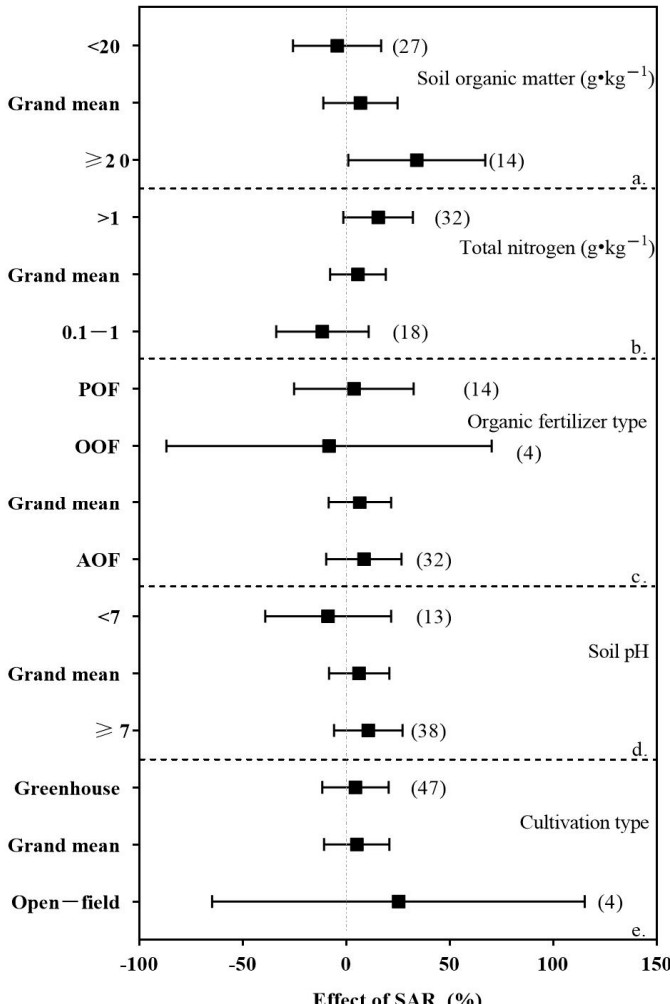

**Figure 7.** Organic fertilizers combined with soil organic matter (**a**), total soil nitrogen (**b**), type of organic fertilizers (**c**), soil pH (**d**), and cultivation type (**e**) on tomato SAR (sugar/acid content ratio) percentage changes. Points represent the mean effect, while bars represent 95% confidence intervals. The numbers on the right side of the confidence intervals represent the sample sizes. AOF, animal organic fertilizer; MOF, mixed animal-and-plant organic fertilizer; POF, plant organic fertilizer; OOF, other organic fertilizer.

*3.6. Effect of Organic Fertilizer Combined with Soil Organic Matter, Total Soil Nitrogen, and Organic-Fertilizer Type on VC*

The organic fertilizers increased tomato VC content when the soil organic matter was < 20 g·kg$^{-1}$, which increased by 10.09%. The soil organic matter was ≥20 g·kg$^{-1}$ and had a significant positive effect on tomato VC content, which increased by 32.02% (Figure 9a). Total soil nitrogen content had no effect on tomato VC rise when organic fertilizers was applied versus no fertilizer application (Figure 9b). Among the organic-fertilizer types, animal organic fertilizer, plant organic fertilizer, and mixed animal-and-plant organic fertilizer all had significant effects on the improvement of tomato VC content, with percentage changes of 16.66%, 20.84%, and 24.45%, respectively; mixed animal and plant organic fertilizer had the most significant effect (Figure 9c). At a soil pH < 7, the increase of organic fertilizers had a significant effect on the increase of tomato VC, increasing by 31.26%, while at a soil pH ≥ 7, the percentage change was 18.84% (Figure 9d). Other organic fertilizers had no significant effect on tomato VC content. The effect of organic fertilizers applied in the open field was significantly higher than that in the greenhouse, with a percentage change of 27.01% and 15.28%, respectively (Figure 9e).

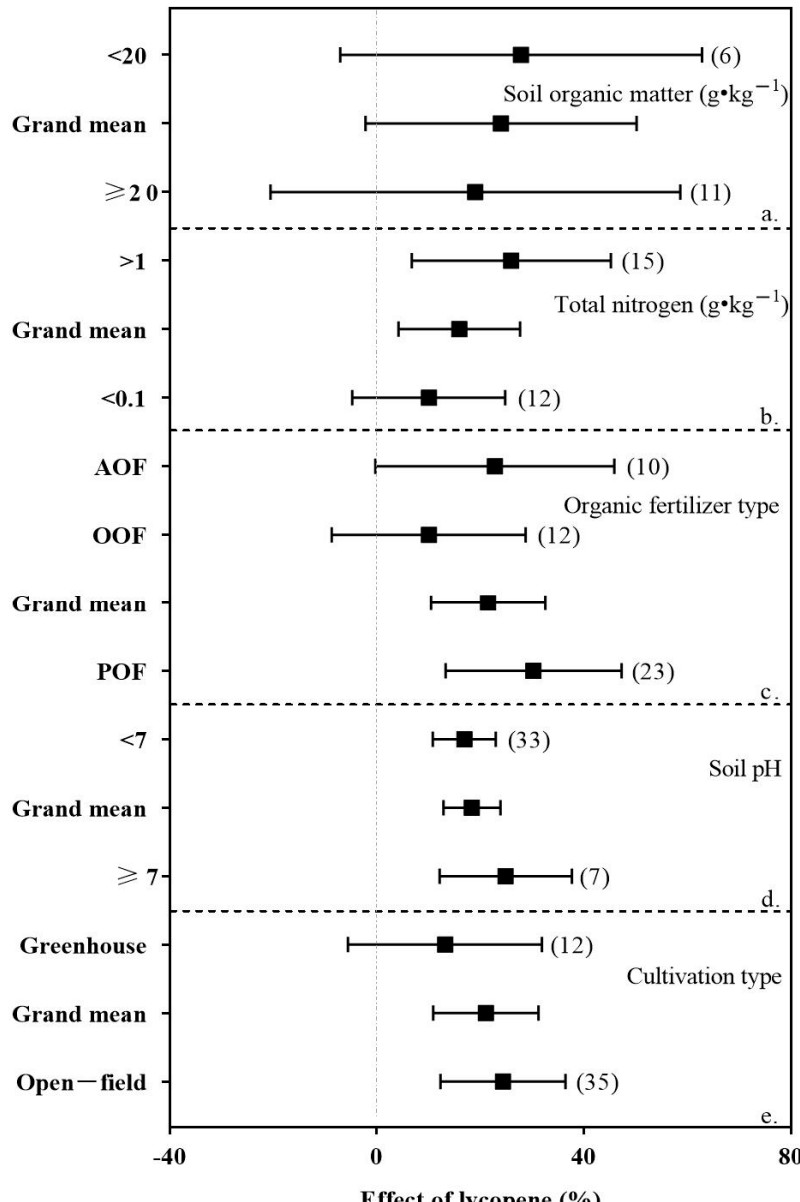

**Figure 8.** Organic fertilizers combined with soil organic matter (**a**), total soil nitrogen (**b**), type of organic fertilizers (**c**), soil pH (**d**), and cultivation type (**e**) on tomato lycopene percentage changes. Points represent the mean effect, while bars represent 95% confidence intervals. The numbers on the right side of the confidence intervals represent the sample sizes. AOF, animal organic fertilizer; MOF, mixed animal-and-plant organic fertilizer; POF, plant organic fertilizer; OOF, other organic fertilizer.

*3.7. Effect of Organic Fertilizer Combined with Soil Organic Matter, Total Soil Nitrogen and Organic-Fertilizer Type on Nitrate*

Except for organic-fertilizer type and cultivation type, soil organic matter, total soil nitrogen, and soil pH had no significant effect on tomato nitrate content under organic fertilizers application (Figure 10a–d). In the subgroup of cultivation type, there was no significant change in the increase of tomato nitrate content with percentage changes of 11.77% and 5.74%, respectively (Figure 10e).

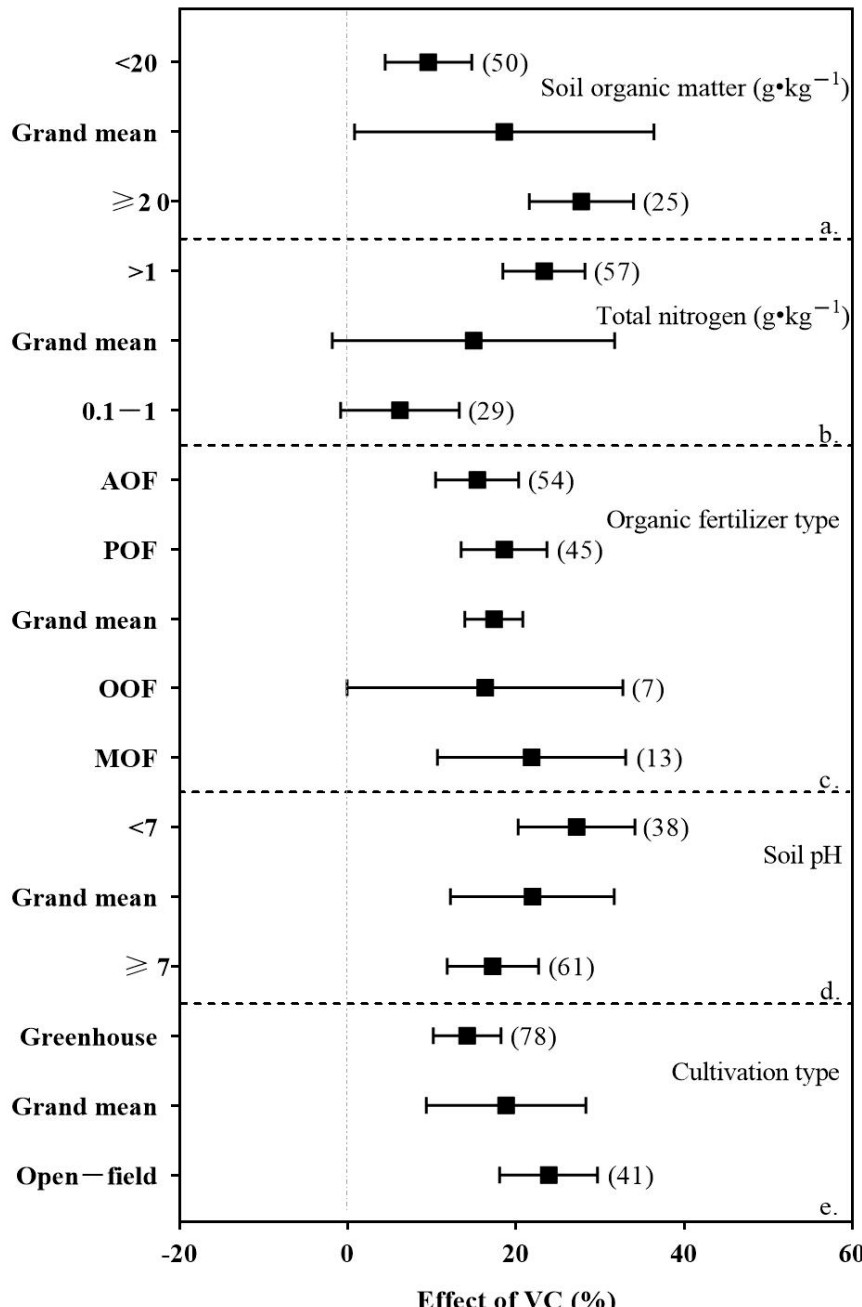

**Figure 9.** Organic fertilizers combined with soil organic matter (**a**), total soil nitrogen (**b**), type of organic fertilizers (**c**), soil pH (**d**), and cultivation type (**e**) on tomato VC (vitamin C) percentage changes. Points represent the mean effect, while bars represent 95% confidence intervals. The numbers on the right side of the confidence intervals represent the sample sizes. AOF, animal organic fertilizer; MOF, mixed animal and plant organic fertilizer; POF, plant organic fertilizer; OOF, other organic fertilizer.

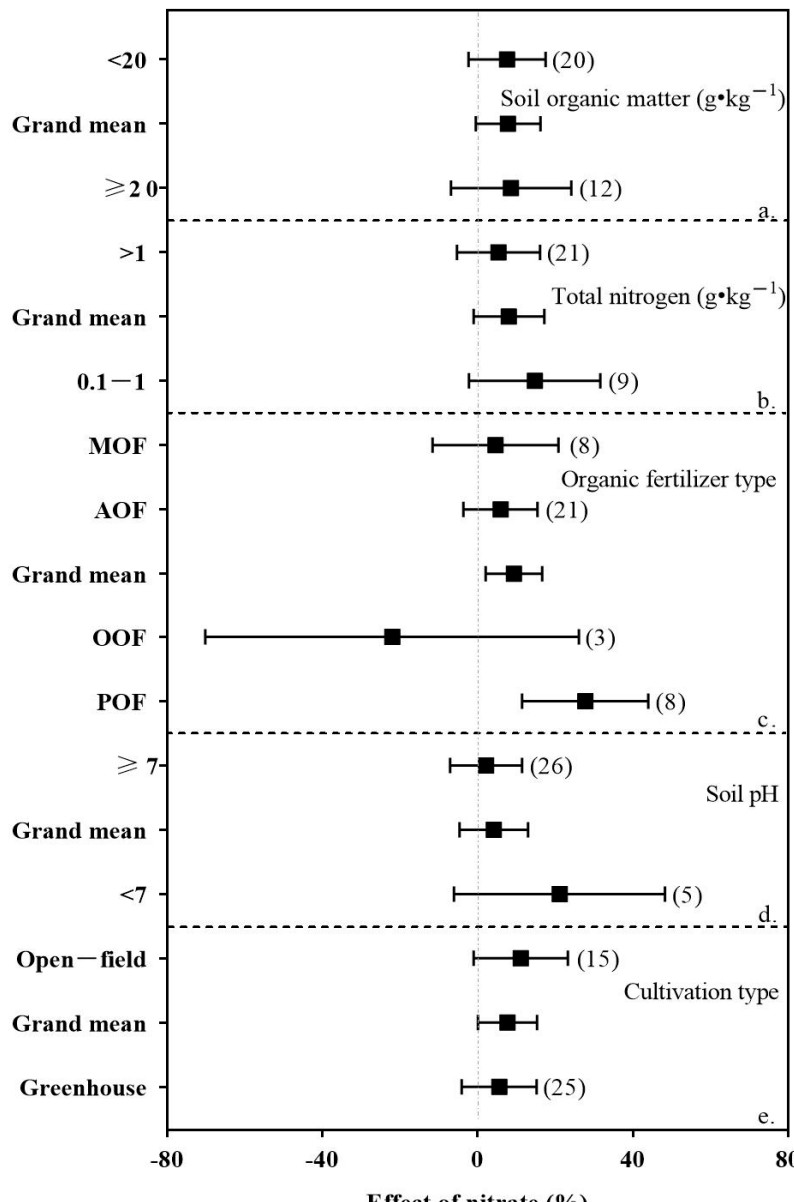

**Figure 10.** Organic fertilizers combined with soil organic matter (**a**), total soil nitrogen (**b**), type of organic fertilizers (**c**), soil pH (**d**), and cultivation type (**e**) on tomato nitrate percentage changes. Points represent the mean effect, while bars represent 95% confidence intervals. The numbers on the right side of the confidence intervals represent the sample sizes. AOF, animal organic fertilizer; MOF, mixed animal-and-plant organic fertilizer; POF, plant organic fertilizer; OOF, other organic fertilizer.

## 4. Discussion

### 4.1. Effect of Organic Fertilizer Combined with Soil Organic Matter, Total Soil Nitrogen, and Organic-Fertilizer Type on Yield

The tomato yield increased with an increase in soil organic matter. The nutrient and water retention capacity increased with organic matter accumulation, thus increasing tomato yield [28]. The figure demonstrates that the tomato yield rose more slowly as the total nitrogen concentration of the soil increased compared to when no fertilizer was applied. It is possible that the nitrogen content exceeded the optimal value due to the high application rate of organic fertilizers, causing tomato yields to increase slowly and sometimes decrease [29]. Animal-derived organic fertilizers were better than other organic fertilizers at increasing the tomato yield, possibly because their nitrogen release rate was

higher than that of other organic fertilizers [30–32]. They also improved soil physical properties and soil nutrient utilization efficiency [33,34]. Soil pH affects the water and mineral nutrient uptake by tomato plants [35]. The figure depicts that the yield percentage change of soil pH < 7 was 1.82 times that of soil pH $\geq$ 7. Adding organic fertilizers resulted in a higher tomato yield under acidic soil conditions than in alkaline soil. Furthermore, fertilizers application changes the soil microbial community structure, especially AOA and AOB involved in the nitrification process. Applying organic fertilizers is conducive to the digestion process, thereby improving Nr and NO3-N, stabilizing NH4-N, reducing $NH_3$ volatilization [36], and further improving tomato yield. All cultivation types had a positive effect on tomato yield effect values, with open-field cultivation producing significant differences in tomato yield compared to facility cultivation, with reduced yield variability in greenhouse production compared to field production systems [37]. This could be due to differences in tomato varieties, growth habits, disease resistance, and other characteristics.

Although organic fertilizers improve tomato yields and quality, they also have some disadvantages in atmospheric and soil pollution compared to conventional cultivation. Studies have shown that methane and carbon dioxide emissions from rice paddies can be increased by the excess application of organic fertilizers. Moreover, improperly disposed food waste can produce bad odors and harbor pathogenic microorganisms that cause environmental pollution [38]. Similarly, soil applied with animal manure may significantly aggravate the heavy metal accumulation in Chinese farm soils coming from Cu and Zn, for example [39]. In recent years, it has been found that animal organic fertilizers could produce large amounts of ammonia, which indirectly or directly has some impact on human health [40,41]. However, in South, Southwest, and Northwest China, pyrolysis gasification technology has been used to degrade domestic waste, which may provide the possibility to ameliorate the problem of environment pollution and boost the sustainable development of agriculture. Therefore, there are still difficulties and challenges in optimizing the application technique of organic fertilizers in order to avoid environmental pollution as much as possible.

### 4.2. Effect of Organic Fertilizer Application on Changes in Tomato Quality under Soil Organic Matter

In terms of soluble solids, soluble sugar, sugar/acid, and VC content, distinct varieties of tomato responded differentially to varying levels of soil organic matter. There was no significant difference in the soluble solids of tomato under different soil-organic-matter contents. There was no significant change in tomato soluble sugar with the addition of organic fertilizers when the soil organic matter was $\geq$20 g·kg$^{-1}$, which was consistent with the previous report that tomato soluble sugar decreased with the increase of fertilizers application. Organic matter positively affected the tomato sugar/acid ratio and VC content when soil organic matter was $\geq$20 g·kg$^{-1}$. Organic matter plays an important role in soil fertility and function. The trace elements in organic matter can meet the requirements of soil microorganisms, promote microbial activities, affect soil–microorganism interaction [42], and indirectly affect crop quality. The application of organic fertilizer positively affected tomato soluble solids under different soil-organic-matter conditions, with no significant difference between them. The tomato sugar acid ratio is an important factor influencing tomato flavor quality, and it increased significantly under the high soil organic matter content of organic fertilizers. Relevant studies have shown that organic fertilizers increase soil organic matter, thereby improving the activity of soil bacteria to break down soil organic matter and release nitrogen, phosphorus, and potassium [43–45], which have a positive effect on soil enzyme activity, increasing the sugar/acid radio [46,47]. When soil organic matter was $\geq$20 g·kg$^{-1}$, the percentage change of organic fertilizers on tomato VC was 3.17 times higher than that of soil organic matter <20 g·kg$^{-1}$. Organic fertilizers had a linear effect on tomato VC elevation as the soil-organic-matter content increased.

### 4.3. Effect of Organic Fertilizer Application on Changes in Tomato Quality under Total Soil Nitrogen

Except for nitrate, increasing total soil nitrogen content with additional organic fertilizers positively affects tomato soluble solids, soluble sugars, lycopene, and VC. However,

as the total soil nitrogen increases, nitrogen absorbed by the roots could reach the peak and lead to an excessive accumulation in the roots upon the reapplication of organic fertilizers; this finally results in a lower nitrate content being provided to the fruit [48]. Meanwhile, excess soil nitrogen may affect the tomato plant's uptake of water and nutrients from the soil [49], ultimately leading to an adverse influence on tomato quality. The addition of organic fertilizers can change soil microbial activity, allowing microorganisms to convert inorganic nitrogen to organic [50], and different organic fertilizers can change soil physicochemical properties [51,52], enhancing tomato growth and quality.

### 4.4. Effect of Different Organic-Fertilizer Types on Changes in Tomato Quality

For the organic-fertilizer types, both animal and plant organic fertilizer increased tomato soluble solids, soluble sugars, and VC content, with no significant change in the sugar/acid ratio, while plant organic fertilizer had a positive effect on lycopene and nitrate content. For example, earthworm manure improves the root vigor of tomato plants by affecting soil ecology, which in turn promotes the growth of seedlings in the early stages of the crop [53], improves the photosynthetic capacity of the leaves, and promotes nutrient uptake and transport, thus ultimately improving tomato yield and quality. Studies have shown that diluting rapeseed cake manure fermentation solution by 20 to 50 times can improve the nitrogen metabolism of tomatoes [54], significantly promote the growth and development of tomato roots, and positively affect tomato yield and quality.

### 4.5. Effect of Organic Fertilizer on Tomato Quality at Different pH and Cultivation Types

Organic fertilizers are essential for tomato growth, and their effect on tomato quality varies at different pH values. At a pH < 7, the increased application of organic fertilizers significantly increased the soluble solids, soluble sugars, lycopene, and VC content. This indicates that organic fertilizers improved fruit quality and increased the concentration of soluble solids, sugars, and vitamin C [55,56]. At a pH $\geq$ 7, the tomato lycopene content was improved more by additional organic fertilizers than at a pH < 7. This difference may be because tomato lycopene content depends not only on the growing conditions but also on the cultivation method and variety [57]. Organic fertilizers contain not only nutrients, but also organic matter required for plants. Organic matter plays an important role in plant growth and development by releasing nutrients, improving soil physical and chemical properties, and promoting root activity.

Among the cultivation types, the greenhouse had less temperature fluctuations than open-field cultivation and higher humidity, which was more favorable for tomato growth. However, the percentage change of organic fertilizers application on tomato VC content in open-field cultivation was 1.77 times higher than the percentage change in the greenhouse. The lack of moisture in open-field cultivation resulted in higher VC content in tomatoes than in the greenhouse. These results are consistent with a previous study that observed that moisture stress can promote VC content [58–61].

### 5. Conclusions

The results demonstrate that using organic fertilizers can improve tomato yield and quality. In general, organic fertilizers' application significantly improved most of the qualities, such as TSS, SS, lycopene, VC, and nitrate content, with no significant change in the sugar/acid ratio. Although the effects of total soil nitrogen on tomato yield and pH on lycopene were significantly different from other variables, other variables had no such difference. TSS, SAR, and VC improved with the increase of soil organic matter. TSS, SS, lycopen, and VC improved when total soil nitrogen was >1 g·kg$^{-1}$. Different organic fertilizers had different effects on tomato yield and quality. Animal organic fertilizer had a significant positive effect on yield, while in terms of quality, the animal-and-plant organic fertilizer had the best improvement effect compared to the other organic fertilizer. The yield and partial quality (TSS and VC) of tomatoes increased significantly with different

cultivation methods and soil pH. The effect of acid soil on yield and quality improvement was significantly higher than that of alkaline soil (i.e., compared to pH > 7).

**Author Contributions:** Conceptualization, L.Y. and F.G.; methodology, F.G.; validation, L.Y., F.G. and Y.Z.; formal analysis, F.G., R.L., Y.Z.; investigation, F.G., H.L., X.M., H.G.; writing—review and editing, F.G.; supervision, K.C. and L.Y.; funding acquisition, L.Y. All authors have read and agreed to the published version of the manuscript.

**Funding:** The study was supported by National Key Research Programme (2021YFD1600300), Key Research Program of Ningxia Hui Autonomous Region of China (2021BBF02019), and Key Research Program of Ningxia Hui Autonomous Region of China (2018BBF02009). Jiangsu Province Key Research and Development Program: Modern Agriculture Project (BE2021379). The National Natural Science Foundation of China (31860575).

**Data Availability Statement:** The authors declare that the data supporting this study are available from the corresponding author upon reasonable request.

**Acknowledgments:** We are thankful to all the researchers whose contributions were used in our study analysis and referenced in this review article.

**Conflicts of Interest:** The authors declare no conflict of interest.

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
