# Peer review of "Effects of Organic Fertilizer Application on Tomato Yield and Quality: A Meta-Analysis"

_applsci, doi:10.3390/app13042184_

Round 1
Reviewer 1 Report
The manuscript entitled ''Effects of Organic Fertilizer Application on Tomato Yield and Quality: A meta-analysis'' emphasized a very positive view of organic fertilizers and their benefits for the environment, soil, and plant, especially tomato.
I think the manuscript was written with a little bias. Of course, the application of organic fertilizers has an important role in order to organic farming and sustainable agriculture. But it doesn't mean that they have not any adverse! With a closer look at some newly published studies, it's understandable that using organic fertilizers with goals of organic farming, despite the increasing quality of the product may doesn't make satisfaction with product yield more than conventional cultivation.
Also, authors considered animal waste as one of the organic fertilizers groups with notable effectiveness in most parameters! But, there is nothing about its disadvantages in soil and environments. Releasing pollution such as heavy metal is one of the side effects of untreated animal waste. Even some household waste needs to be modified before being reused in agriculture. But there is nothing about the disadvantages or challenges with organic fertilizers in current work!
It is highly suggested to add publication bias tests like Egger tests. There is just an example you can use: https://doi.org/10.1002/ldr.4464
There are some small points in the PDF.

Author Response
对审核人一意见的回应
Dear Reviewer,
Thank you very much for allowing us to revise and resubmit this manuscript.
Below we have provided a detailed response to your comments, concerns, suggestions, etc.
We have indicated the changes made in response to the reviewers' comments in the body of the revised manuscript.
We have made efforts to ensure the revised manuscript is improved precisely according to the Reviewers' comments.
Reviewer #1: The manuscript entitled ''Effects of Organic Fertilizer Application on Tomato Yield and Quality: A meta-analysis'' emphasized a very positive view of organic fertilizers and their benefits for the environment, soil, and plant, especially tomato.
Point 1: I think the manuscript was written with a little bias. Of course, the application of organic fertilizers has an important role in order to organic farming and sustainable agriculture. But it doesn't mean that they have not any adverse! With a closer look at some newly published studies, it's understandable that using organic fertilizers with goals of organic farming, despite the increasing quality of the product may doesn't make satisfaction with product yield more than conventional cultivation.
Response 1: Thank you very much for this valuable comment. We agree with this comment. We have reviewed the literature and added the disadvantages of organic fertilizers. Thanks again for your valuable advice. Please see lines 556-561 for details.
Point 2: Also, authors considered animal waste as one of the organic fertilizers groups with notable effectiveness in most parameters! But, there is nothing about its disadvantages in soil and environments. Releasing pollution such as heavy metal is one of the side effects of untreated animal waste. Even some household waste needs to be modified before being reused in agriculture. But there is nothing about the disadvantages or challenges with organic fertilizers in current work!
Response 2: Thank you very much for this valuable comment. We agree with this comment. We add to the literature on heavy metal contamination of soils and environmental pollution by organic fertilizers and the challenges in the future. Please see lines 561-570 for details.
Point 3: It is highly suggested to add publication bias tests like Egger tests.
Response 3: Thank you very much for this valuable comment. We have added publication bias detection for Egger tests as you suggested. Please see lines 246-247 and Table 1 for details.
Point 4: There are some small points in the PDF.
Response 4: Thank you very much for this valuable comment. We have modified small points in the PDF. Please see lines 20-29; 36-41;148-149; 272-273; 317-320 and lines 549 for details.

Reviewer 2 Report
Overall, the manuscript is in good condition, however it does not read well. As a result, I recommend that the manuscript be published with some minor adjustments that would improve it even more. Please see my line-by-line comments and suggestions, particularly writing and grammatical suggestions, before publication.
My suggestions for authors are listed below:
You can use my suggested paragraph or one that is more fluid in the abstract:
Abstract:
Tomatoes are a globally cultivated and popular vegetable. The output and quality of tomatoes are significantly influenced by the use of organic fertilizers. It was discovered that organic fertilizers increase tomato productivity and improve fruit quality. The influence of organic fertilizers on tomato yield and quality is shown to be complex and dependent on soil organic matter, total soil nitrogen, organic fertilizer kinds, and other variables. In this review paper, we evaluated 769 data sets from 107 research papers and determined that organic fertilizers can enhance tomato yield by 42.18 %. Compared to the control group, soluble solids, soluble sugar, lycopene, vitamin C, and nitrate were raised by 11.86%, 42.18%, 23.95%, 18.97%, and 8.36%, respectively. In general, soil organic matter > 20 g/kg and organic fertilizer significantly improved the tomato sugar/acid content ratio and VC, whereas under total soil nitrogen > 1 g/kg, organic fertilizer had significant differences in tomato soluble solids, soluble sugar, lycopene, and vitamin C, with different organic fertilizer types having different effects on tomato quality. Comparing animal and plant organic fertilizers to other forms of organic fertilizers, tomato quality varied significantly. We also evaluated the impact of different cultivation methods, soil organic matter, total soil nitrogen, soil pH, and types of organic fertilizer on tomato yield and quality. The results gave valuable information and direction for the use of organic fertilizers in greenhouse production.
Introduction:
Line 39: ..anti-cancer
Line 41: are produced worldwide,
Line 44: …is currently one of the most commonly used methods
Line 47-48: more than 50% of the nitrogen and 90% of the phosphorus
Line 48: water sources [9],
Line 50-51: Furthermore, excessive chemical fertilizer application can result in decreased food safety and lower vegetable quality, such as nitrate accumulation in plants [13].
Line 54-55: Organic fertilizer has a low nutrient..
Line 55: depends on the water and temperature conditions of the soil.
Line 61: and the titratable acid
Line 62: fertilizer utilization rates and reduce
Line 63: reducing environmental pollution
Line 64: on the qualitative analysis of
Line 65: only a few studies have investigated the
Line 67-68: and quality, with
Materials and methods:
Line 72-74: A complete literature search was undertaken on ISI Web of Science, China National Knowledge Internet, and Google academic to collect data for a study on the effects of organic fertilizer on tomato output and quality published as of April 2022.
Line 77: studies: (1) The same
Line 78-80: (2) describe the test site, cultivation type, and organic fertilizer type; (3) provide the mean and standard deviation of the treatment group and the control group, and N (number of replicates).
Line 81: SD using the formula S? = ??×√? [25].
Line 82-83: ..soluble solids (TSS), soluble sugar (SS), sugar/acid content ratio (SAR), lycopene, vitamin C (VC), and nitrates.
Line 84: ..organic fertilizer type, and cultivation type as subgroups.
Line 89:…. (1) Animal (insert space before animal) ….. organic fertilizer, and (4) other
Line 91: ..769 data pairs were analyzed to determine the effect ..
Line 93: … is displayed in Figure 1.
Line 95-96: 8.0 software. Using the
Line 101: Where t deals with the average of the set of variables, c is the average of the control variables, lnR is the logarithmic response, and the variance…
Line 104: where ?? is
Line 105: …interpretation, the formula percentage…
Line 106-107: heterogeneity and indicate differences
Line 117: ..respectively),
Results:
Line 121-122: This is not a sentence.
Line 125-126: SS by 42.18% (95% CI: 25.51% to 61.06%), and SAR by 6.17%
Line 126: SAR indices of..
Line 128: …22.80%), and
Line 147-148:.. has no effect on tomato yield…
Line 149: 0.1–1 g/kg
Line 169-170:… ≥ 20 g/kg and < 20 g/kg, the tomato’s soluble solid content increased by 14.17 and 11.47%, respectively…
Line 171: ..1g/kg and 0.1–1 g/kg, …
Line 174:.. improving tomato soluble solids tended ..
Line 175: on improving tomato solubles (Figure 5c)…
Line 176-177: …organic fertilizer, animal organic fertilizer, and mixed organic fertilizer, respectively.
Line 179: …whereas at pH > 7,
Line 190-191: When organic matter was 20 g/kg, the percentage change of organic fertilizer on the soluble sugar of tomato increased by 60.35 % (Figure 6a).
Line 193: 0.1–1 g/kg
Line 193: had positive effects on
Line 197-199: However, there was no significant difference between mixed organic and other types of organic fertilizers in the percentage changes in tomato soluble sugar.
Line 199:.. intervals overlapped, and the percentage changes were
Line 203-204: Both cultivation types had positive effects on the soluble sugars.
Line 209-210: ..under increased application of..
Line 222-223: When total soil nitrogen was > 1 g/kg, the lycopene increased by 29.67%, while when it was…
Line 225-226: ..had no significant effect on tomato lycopene..
Line 245: 10.09%. Soil organic
Line 246: content, which increased by..
Line 246-247 : Total soil nitrogen content had no effect on tomato VC rise when organic fertilizer was applied versus no fertilizer application (Figure 9b).
Line 251: 20.84, and
Line 253-254: significant effect on the increase of tomato VC, increasing by 31.26%,
Line 257: with a percentage change of 27.01 and 15.28%, respectively
Line 268: (Figure 10a–d).
Line 268-270: In the subgroup of cultivation type, there was no significant change in the increase of tomato nitrate content with percentage
Discussion:
Line 279: …demonstrates that tomato yield rose more slowly as the total nitrogen concentration of the soil increased compared to when no fertilizer was applied.
Line 282: causing tomato yields to increase slowly and
Line 283: were better than other organic fertilizers at increasing tomato yield,
Line 285-286: They also improved soil physical properties and soil nutrient utilization efficiency [33, 34].
Line 292: , stabilizing
Line 297: resistance, and other characteristics.
Line 301-302: In terms of soluble solids, soluble sugar, sugar/acid, and VC content, distinct varieties of tomato responded differentially to varying levels of soil organic matter.
Line303: difference in the soluble solids of tomato under different soil organic matter contents.
Line 306: increase in fertilizer
Line 307: Organic matter positively affected
Line 311-312: different soil organic matter conditions, with no significant difference between them.
Conclusion:
Line 367: …and nitrate content, with no significant
Line 371: soil nitrogen was > 1 g/kg.
Line 374: to other
Author Response
对审核者2意见的回应
Dear Reviewer,
Thank you very much for allowing us to revise and resubmit this manuscript.
Below we have provided a detailed response to your comments, concerns, suggestions, etc.
We have indicated the changes made in response to the reviewers' comments in the body of the revised manuscript.
We have made efforts to ensure the revised manuscript is improved precisely according to the Reviewers' comments.
Reviewer #2: Overall, the manuscript is in good condition, however it does not read well. As a result, I recommend that the manuscript be published with some minor adjustments that would improve it even more. Please see my line-by-line comments and suggestions, particularly writing and grammatical suggestions, before publication.
Point 1: Grammar and advice on abstract.
Response 1: Thank you very much for this valuable comment. We have taken your suggestions and revised the summary section to make the paragraphs more fluid. Please see lines 12-29 for details.
Point 2: Grammar and advice on introduction.
Response 2: Thank you very much for this valuable comment. We have taken your suggestions and revised the summary section to make the paragraphs more fluid. Please see lines 38; 40; 116; 119-121; 122-123; 127-128; 132-134; 135-136 and lines 139 for details.
Point 3: Grammar and advice on materials and methods.
Response 3: Thank you very much for this valuable comment. We have taken your suggestions and revised the summary section to make the paragraphs more fluid. Please see lines 144-146; 148-153; 154-155; 156; 161-162 ;164-166; 233; 238-239 and 241-244 for details.
Point 4: Grammar and advice on results.
Response 4: Thank you very much for this valuable comment. We have taken your suggestions and revised the summary section to make the paragraphs more fluid. Please see lines for 275-276; 319-320; 353-356; 358-359; 361; 363-363; 388-390; 402-404; 407; 414-415; 441-443; 445-446; 467-469; 472-474; 492-493 and 507-508 for details.
Point 5: Grammar and advice on discussion.
Response 5: Thank you very much for this valuable comment. We have taken your suggestions and revised the summary section to make the paragraphs more fluid. Please see lines 527-528; 530-534; 577-580; 583; 585-586 and 588-589 for details.
Point 6: Grammar and advice on conclusion.
Response 6: Thank you very much for this valuable comment. We have taken your suggestions and revised the summary section to make the paragraphs more fluid. Please see lines 664 and 668 for details.

Reviewer 3 Report
In general a very good and organized review manuscript.
I have few suggestions as follow:
Keywords: Please, substitute the keywords. All of them are presented in the title. Suggestion: organic fertilization; lycopen; nitrogen...
Figure 1: Please, confirm the Brazil abbreviation in the map. Also, I suggest that the names of the countries should be described.
Figure 1. Geographic distribution of the experimental sites in this meta-analysis. CAN: Canada; USA: United States of America; MEX: Mexico; BRA: Brazil...
Figures 2; 4-10: Do the numbers in parentheses are the number of samples? The same for figures 4-10.
Line 136: organic fertilizer. I suggest that because the authors used this form in the text. Only here in the figure 2 title the word fertilizer appear with "s" (fertiliser).
Some curves in Figure 3 needs to be addressed with r2 value and p-value.
Figures in general: Points represent the mean effect, while bars represent 95% confidence intervals?Please confirm this information on all other figures.
Other figures must also contain this information: "AOF: animal organic fertilizers, MOF: mixed animal and plant organic fertilizers, POF: 164 plant organic fertilizers, OOF: other organic fertilizers."
Line 186: TSS (Total soluble solids)
Line 214: SS (soluble sugar)
Line 217: SAR (sugar/acid content ratio).
Line 260: VC (vitamin C)
Section 4.3.Why is nitrate an exception? This could be discussed.
Why is nitrate an exception? This could be discussed.
Author Response
对审核者3意见的回应
Dear Reviewer,
Thank you very much for allowing us to revise and resubmit this manuscript.
Below we have provided a detailed response to your comments, concerns, suggestions, etc.
We have indicated the changes made in response to the reviewers' comments in the body of the revised manuscript.
We have made efforts to ensure the revised manuscript is improved precisely according to the Reviewers' comments.
Reviewer #3: In general a very good and organized review manuscript.I have few suggestions as follow:
Point 1: Keywords: Please, substitute the keywords. All of them are presented in the title. Suggestion: organic fertilization; lycopen; nitrogen...
Response 1: Thank you very much for this valuable comment. We agree with this comment. We have reworked the keywords as you suggested. Please see lines 30.
Point 2: Figure 1: Please, confirm the Brazil abbreviation in the map. Also, I suggest that the names of the countries should be described. Figure 1. Geographic distribution of the experimental sites in this meta-analysis. CAN: Canada; USA: United States of America; MEX: Mexico; BRA: Brazil...
Response 2: Thank you very much for this valuable comment. We have confirmed the abbreviation Brazil and added the name of each country to the chart. Please see Figure 1.
Point 3: Figures 2; 4-10: Do the numbers in parentheses are the number of samples? The same for figures 4-10.
Response 3: Thank you very much for this valuable comment. Since we did not express it clearly, very sorry for your misunderstanding. The numbers of Figure 4-10 are the number of samples. Please see lines 300-301; 347-347; 381-382; 422-423; 434-435; 459-460; 499-450 and 518-519 for details.
Point 4: Line 136: organic fertilizer. I suggest that because the authors used this form in the text. Only here in the figure 2 title the word fertilizer appear with "s" (fertiliser).
Response 4: Thank you very much for this valuable comment. We have changed the fertiliser to a fertilizer. Please see lines 299 for details.
Point 5: Some curves in Figure 3 needs to be addressed with r2 value and p-value.
Response 5: Thank you very much for this valuable comment. We agree with this comment. We have added R2 and p as your suggestion. Please see Figure 3.
Point 6: Figures in general: Points represent the mean effect, while bars represent 95% confidence intervals? Please confirm this information on all other figures.
Response 6: Thank you very much for this valuable comment. We agree with this comment. We have confirmed points represent the mean effect and bars represent 95% confidence intervals on all figures. Please see lines 300-301; 347-347; 381-382; 422-423; 434-435; 459-460; 499-450 and 518-519 for details.
Point 7: Other figures must also contain this information: "AOF: animal organic fertilizers, MOF: mixed animal and plant organic fertilizers, POF: 164 plant organic fertilizers, OOF: other organic fertilizers."
Response 7: Thank you very much for this valuable comment. We agree with this comment. We have added the name of the organic fertilizers to the figures as you suggested. Please see Figure 2; 4-10.
Point 8: Line 186: TSS (Total soluble solids); Line 214: SS (soluble sugar); Line 217: SAR (sugar/acid content ratio); Line 260: VC (vitamin C).
Response 8: Thank you very much for this valuable comment. We agree with this comment. We have completed total soluble solids, soluble sugar, sugar/acid content ratio and vitamin C as you suggested. Please see lines 380; 421; 433 and 498 for details.
Point 9: Section 4.3.Why is nitrate an exception? This could be discussed.
Response 9: Thank you very much for this valuable comment. We agree with this comment. We have reviewed the literature to add the nitrate component not discussed. Please see lines 614-618 section 4.3.

Round 2
Reviewer 1 Report
The authors did some corrections. It potentially can be considered for publication.